# Effects of Delayed Mating on the Reproductive Performance of *Henosepilachna vigintioctopunctata* (F.) (Coleoptera: Coccinellidae)

**DOI:** 10.3390/insects12070629

**Published:** 2021-07-10

**Authors:** Ya-Ling Wang, Qi-Nian Jin, Xiang-Ping Wang

**Affiliations:** 1College of Agriculture, Yangtze University, Jingzhou 434025, China; 18209364270@163.com (Y.-L.W.); jinqinian@gmail.com (Q.-N.J.); 2Forewarning and Management of Agricultural and Forestry Pests, Hubei Engineering Technology Center, Yangtze University, Jingzhou 434025, China

**Keywords:** *Henosepilachna vigintioctopunctata*, fecundity, fertility, mating delay, population growth parameters

## Abstract

**Simple Summary:**

In many Asian countries, *Henosepilachna vigintioctopunctata* (F.), is seriously harmful to Solanaceae vegetables. With the popularization of green agriculture and the improvement in people’s living standards, biological pest control may become the mainstream. The artificial release of sex pheromones and other methods to delay insect mating, thus affecting population abundance, is an important part of biological control. We took *H. vigintioctopunctata* collected from Jingzhou, Hubei Province, China, back to the laboratory to establish an experimental population to study the effect of delayed mating on its reproductive behavior. The negative effects on reproduction and changes in population life table parameters, such as net reproductive rate, intrinsic and finite rates of increase, doubling time, and mean generation time, could be estimated by the treatment of delayed mating of males and females, which could be useful for providing important information for pest control in the future.

**Abstract:**

*Henosepilachna vigintioctopunctata* (F.) is a serious pest of numerous solanaceous crops in many Asian countries. The purpose of this study was to clarify the effects of delayed mating on mating success, fecundity, fertility, pre-oviposition period, oviposition period, adult longevity, and population life table parameters (including net reproductive rate, intrinsic and finite rates of increase, doubling time, and mean generation time) of *H. vigintioctopunctata*. Beginning three days after emergence for both sexes, mating was delayed an additional 0, 2, 4, 6, or 8 days. We compared the data when mating was delayed for males only with the data when mating was similarly delayed for females only. Reproductive and life table parameters were calculated from the two data sets and compared. The results showed that the preoviposition and oviposition period of adults was significantly reduced by delayed mating, while the preoviposition period was not significantly different in adults mated at older ages. The mating success rate, fecundity, and proportion of hatching eggs decreased with increasing mating age. Longevity was not affected by the age at mating. Mating delay also affected the life table parameters of *H. vigintioctopunctata*, with a similar trend observed in the net reproductive rate and intrinsic and finite rates of increase, all of which decreased gradually as the number of delay days increased. The population doubling time increased with increases in mating age. The results also showed that delayed mating was an effective measure to consider in controlling *H. vigintioctopunctata*. It is hoped that our data will provide a scientific basis and contribute technical guidance for forecasting and integrated management of this pest.

## 1. Introduction

The Hadda beetle, *Henosepilachna vigintioctopunctata* (Coleoptera: Coccinellidae), is widely distributed in Australia [1], Brazil [1], China [2,3], Japan [4], and other Asian countries [5,6]. Both larvae and adults feed mainly on the leaves of Solanaceae plants [7], including *Solanum nigrum* L., *S. tuberosum* L. [8], and *Lycopersicon esculentum* Mill. [5]. Initially, the newly hatched larvae aggregate mainly on the back of the leaves to feed on mesophyll [9]. In later instars, the damage they cause becomes serious, with the affected leaves usually forming irregular transparent spots or perforations, which eventually turn into brown concave spots [1,9,10]. Young stems, petals, sepals, and fruits are also damaged by the larvae [10,11], resulting in a reduction in plant yield, poor fruit quality, and even plant death, all of which seriously affect the economic value of the affected crop [12,13,14,15]. The beetles are highly adaptable to temperature variations and have a prolific breeding ability, with 5–6 generations a year in the Jianghan Plain [2,16]. Field investigation showed that the abundance of *H. vigintioctopunctata* varied greatly on different host plant species, and the beetles have the highest population density on *Solanum tuberosum* L. and *Solanum melongena* L. [2,16,17]. Moreover, on the same host plant species, the density of larvae and adults are also quite different [2,17]. Especially when the adults lay eggs, they have a very strong selectivity to the host plant species, and the previous research from our laboratory found that the host plant that the adults like to feed on may not be the same as the host plant on which they choose to lay eggs [18]. These variables are often responsible for the difficult task of controlling the pest.

Even though the management techniques used against *H. vigintioctopunctata* are varied, their efficacy is often limited [19]. The efficiency of artificial removal of egg mass and the use of adult suspended animation to capture adults is low. The effect of control using microbial insecticides (a *Beauveria bassiana* wettable powder, *Beauveria bassiana* oil suspension agent, and engineering bacteria preparations for Coleoptera beetles) is slow [20], and the application of the fungus should be determined according to the meteorological conditions and the actual growth of the crops [21], so pathogenic control has certain limitations. Chemical control is the most common method used to suppress the beetles [22,23]. However, because their eggs and larvae are usually found on the undersides of the leaves, their direct contact with insecticides is often limited [24,25]. In addition, when chemical controls are used, the effective period of control is frequently short term. Repeated sprays are often necessary, increasing the cost of control, and often adversely affecting nontargeted organisms, including beneficial species. None of these factors are conducive to the sustainable development of agricultural ecosystems [26]. There is a growing need for alternative, effective, non-chemical control methods to manage economically important insects [27,28,29,30]. There is a critical need to devise suitable means for replacing chemical control in order to further improve the production and quality of agricultural products, as well as to foster an awareness of providing environmental protection.

Reproduction rate is an important physiological consideration that affects the population growth of a species. Because mating and oviposition are the two most important behaviors related to reproduction, the age at first mating is an important factor influencing mate preference and fitness outcome in insects [31,32,33]. However, age-dependent mating success and fitness consequences can be highly variable in different species. The mating success rate of *Zygogramma bicolorata* Pallister (Coleoptera: Chrysomelidae) increased with an increase in adult age, while the fecundity and longevity decreased with age. However, only the hatching rate of the eggs was affected when the mating age of males increased [34]. Hata et al. [33] found that delayed mating significantly lowered mating success in *Dasylepida ishigakiensis* (Niijima et Kinoshita) (Coleoptera: Scarabaeidae), and it apparently resulted in reduced reproductive outputs. In the scarab beetle, *Anomala orientalis* (Waterhouse) (Coleoptera: Scarabaeidae), delaying mating for more than 6 days resulted in an approximate 35–50% decrease in fecundity [35]. Gerken and Campbell’s [36] results showed that the number of eggs produced by *Trogoderma variabile* (Coleoptera: Dermestidae) and *T*. *inclusum* (Coleoptera: Dermestidae) was reduced after their initial mating was delayed for 10 and 15 days, respectively. In order to devise effective strategies to control *H. vigintioctopunctata,* it is necessary to have a thorough understanding of its reproductive biology and behavior, in particular its mating and egg production capacity, as well as the factors that may influence these processes [37,38].

A delay in mating will often affect the reproductive physiology and lead to decreased production and adaptability of an insect’s progeny [39]. This phenomenon has been shown to exist in many species [35,40,41,42,43]. Due to the effects generated by sex pheromones, both males and females have readily identifiable targets for their selection of mates [32,33,44,45,46,47]. As a result, most mating can be prevented by pheromone-mediated mating disruption [48,49,50,51,52]. Ideally, mating ceases to occur if a program aimed at mating disruption is completely effective. If mating disruption affects only a percentage of the females within a population, it can still be useful because the fecundity of the population, as a whole, may be substantially affected due to mating delay [27,35,53]. Using mating disruption in commercial enterprises, including fruit orchards and tree nut groves, such as walnut orchards, can potentially control many insect pests [41,54,55].

To evaluate the influence of mating disruption on *H. vigintioctopunctata* populations, a thorough understanding of its reproductive biology is required. Laboratory experiments were conducted to investigate the effects of delays in mating on the reproductive and demographic parameters of the coccinellid beetle, *H. vigintioctopunctata*.

## 2. Materials and Methods

### 2.1. Collection and Rearing of Beetles

More than 200 larvae of *H. vigintioctopunctata* were collected from fields in Jingzhou (112°15′ E, 31°26′ N), Hubei Province, China, in March 2019. Larvae and adults were cultured on *S. nigrum* leaves in an artificial climate chamber (RXZ-280, Ningbo Jiangnan Instrument Factory, Ningbo, Zhejiang Province, China) for two generations. The third generation in the lab was used for the experiment (F3). Newly hatched larvae were fed with fresh *S. nigrum* leaves in plastic culture dishes (100 mm × 17 mm, diameter × height). The larval development, pupation, and emergence of the adults were recorded daily. Adults were separated by sex (the female has a longitudinal indentation at the end of the abdomen) and held in separate containers in the rearing chamber (150 mm × 25 mm, diameter × height) for use in the mating experiments. The experiments were carried out at 26 ± 0.5 °C, 70 ± 5% RH, and a 16L:8D photoperiod.

### 2.2. Mating Experiments and Oviposition Assessment

After adult emergence, the beetles were separated by sex and raised separately in culture glass dishes (150 mm × 25 mm, diameter × height), and the emergence time was marked. Because the adult females and males did not become sexually active until 3 days of age (unpublished data), the experimental mating period began when the adults reached 3 days old. A “0-day-delay” beetle or “3-day-old” is defined as when adult beetles were paired on the third day of emergence. A “2-day-delay” is defined as when beetles were paired on the fifth day after emergence, etc. Beetles were paired within the last 2 h of the photo phase.

We set up the following ten treatment combinations: five female-delayed days (0, 2, 4, 6, and 8 d) paired with 0 d delayed males; and five male-delayed days (0, 2, 4, 6, and 8 d) paired with 0 d delayed females. Virgin beetles of different ages were randomly selected from the corresponding age groups. Males and females were paired in transparent plastic Petri dishes (90 mm × 16 mm, diameter × height). Combined with the previous work experience in the laboratory, the Hadda beetle needs at least 30 min of mating to produce fertile eggs. Therefore, we regarded the mating duration of more than 30 min as the completion of mating in this study. The entire copulation process was monitored and the beetles were separated immediately after mating; the females were transferred to oviposition Petri dishes (90 mm × 16 mm, diameter × height). The oviposition cages were provided with a water-soaked cotton ball to maintain humidity and a fresh *S. nigrum* leaf for food. A liner of filter paper was supplied as an oviposition substrate. The paper and leaves were replaced and the egg mass on them were collected daily. The daily egg production of the female was determined under a microscope. After the eggs hatched, the number of larvae or non-hatched eggs was counted to obtain the corresponding hatching amount. The number of eggs laid, egg hatch rate, female oviposition period, and adult longevity were recorded. Fifteen pairs of successful mating beetles were observed in each treatment.

### 2.3. Statistical Analysis

The data on the number of eggs laid and percentage of eggs hatched, as well as the lengths of the pre-oviposition period, oviposition period, and adult longevity were compared among treatments using a generalized linear model, setting normal and binominal distributions for numerical and percentage data, respectively. Pearson’s correlation was performed to examine the correlation between delayed days of mating and the observed variables. All of the data were analyzed using the computer program DPS 6.55 [56]. The figures were made using the Origin Pro.

The life tables of females mating at different ages were calculated according to the age-specific fecundity (*m_x_*) and survival (*l_x_*) rates of females in the different treatments. The following population growth parameters were estimated [57,58]:

Net reproductive rate: $R_{0}=\sum \left(l_{x}m_{x}\right)$; 

The intrinsic rate of increase (*r_m_*) was obtained by the iterative solution of the equation: $$\sum e^{-r_{m}x}l_{x}m_{x}=1;$$

Finite rate of increase: $\lambda =e^{r_{m}}$; 

Mean generation time: $T_{G}=InR_{0}/r_{m}$; 

Population doubling time: $DT=In2/r_{m}$.

## 3. Results

### 3.1. Mating Success

It was determined that mating delays adversely influenced the mating success in *H. vigintioctopunctata*. The successful copulation rate was the lowest when mating was delayed for 8 days, causing a decrease of 25–32%. Regardless of whether the female or the male delayed mating, the mating success rate gradually decreased as the mating delay increased from 2, 4, to 6 days (Table 1).

### 3.2. Pre-Oviposition Period and Oviposition Period

The pre-oviposition period was 13 days long when mating was not delayed, but decreased as the males and females aged, lasting only 4.50 days when mating by the females was delayed for 8 days (Table 1). There was a negative relationship between the length of mating delay and pre-oviposition period (number of days from after mating to the first egg laid) (F_4,25_ = 23.07, *p* < 0.001 for female-delayed mating; F_4,25_ = 91.66, *p* < 0.001 for male-delayed mating).

The oviposition period without delayed mating was 94.33 d. When females were delayed, the oviposition period was significantly shortened, with the shortest oviposition period of only 45.3 ± 3 d occurring after 8 d of mating delay (Table 1). When the males delayed mating, the oviposition period of the females was significantly shortened when the mating was delayed for 2 days. The oviposition period was significantly decreased with the increase of mating delay days (F_4,25_ = 10.78, *p* < 0.001 for female-delayed mating; F_4,25_ = 10.78, *p* < 0.001 for male-delayed mating).

### 3.3. Longevity of Females and Males

There was a significant difference in life expectancy between the sexes. Overall, males had a shorter lifespan (94.83 d) than females (138.17 d) (Figure 1). Unmated females had significantly greater longevity (162.34 d) than mated females. However, the longevity of both females and males was not obviously affected by mating delay (female-delayed mating: F_4,25_ = 1.51, *p* = 0.230 for females; F_4,25_ = 0.17, *p* = 0.954 for male; male-delayed mating: F_4,25_ = 0.19, *p* = 0.940 for females; F_4,25_ = 0.20, *p* = 0.337 for males).

### 3.4. Reproductive Output and Egg Fertility

The number of days of mating delay negatively affected the fecundity (F_4,25_ = 33.59, *p* < 0.001 for female-delayed mating; F_4,25_ = 33.87, *p* < 0.001 for male-delayed mating; Figure 2). The number of eggs laid was significantly reduced in females that were paired with males that had their mating delayed. When mating was delayed in the females, their progeny production decreased significantly after 2 and 4 d of delay compared to females with 0 d of delay. The number of eggs laid also decreased after 2 and 4 d of delay but stabilized when mating was delayed for 6 d. Fecundity decreased by 50% at 8 days of delay, with only 506.50 eggs. When the males delayed mating, the highest progeny production was at 0 days delay, with an insignificant change in fecundity occurring after a delay of 2 days. Subsequently, the fecundity decreased gradually with an increasing number of delay days.

Mating delay not only significantly reduced the fecundity of *H. vigintioctopunctata*, but also reduced the proportion of hatching eggs in their offspring, especially in the female-delayed mating treatment (F_4,25_ = 52.36, *p* < 0.001 for female-delayed mating; F_4,25_ = 43.19, *p* < 0.001 for male-delayed mating; Figure 3). The results of both treatments showed that a significantly lower hatching rate occurred in adults when their mating was delayed for 6 d (47.45% for female-delayed mating; 50.72% for male-delayed mating).

### 3.5. Correlations between the Reproductive Variables and Age at Mating

The mating, fecundity, hatch rate, pre-oviposition period, and oviposition period percentages were negatively related to mating delay days (Table 2). In both mating treatments, the longevity of both females and males were not correlated with age at mating.

### 3.6. Life Tables and Population Growth Parameters

Delayed mating not only affected the reproductive traits of *H. vigintioctopunctata,* but also affected the life table parameters of the population (Table 3 and Table 4). Compared with the non-delayed mating treatment, delayed mating of females or males resulted in a lower net reproductive rate (R_0_: F_4,25_ = 96.75, *p* < 0.001 for female-delayed mating; F_4,25_ = 109.56, *p* < 0.001 for male-delayed mating). The percentage of mating, fecundity, hatch rate, pre-oviposition period, and oviposition period were negatively related to mating delay days (r_m_: F_4,25_ = 11.63, *p* < 0.001 for female-delayed mating; F_4,25_ = 6.36, *p* = 0.0011 for male-delayed mating; λ: F_4,25_ = 11.58, *p* < 0.001 for female-delayed mating; F_4,25_ = 6.32, *p* = 0.0012 for male-delayed mating). The mean generation time ranged from 146.83 to 165.33 days when females delayed mating for 0–8 days, and there was no significant difference among the treatments (T: F_4,25_ = 1.507, *p* = 0.2304). No significant differences, however, were noted in the mean generation time at different age stages when the male mating was delayed (T: F_4,25_ = 0.194, *p* = 0.9395). The population doubling time was extended with an increase in the mating adult age (D: F_4,25_ = 16.15, *p* < 0.001 for female-delayed mating; F_4,25_ = 8.49, *p* < 0.001 for male-delayed mating). When the females delayed mating or when the males delayed mating, the longest time for the population to be doubled was at an age of 11 days.

The population growth parameters were positively correlated with the mating delay days. There was also a positive correlation between the population doubling time and mating delay days. Net reproductive rate, intrinsic rate of increase, and finite rate of increase were negatively correlated with the mating delay days.

## 4. Discussion

Previous studies have documented that the age of mating is a crucial factor in the mating success of female insects [43,59]. Our results showed that a 25–32% reduction in the successful mating rate of *H. vigintioctopunctata* was caused by an 8-day delay in mating, irrespective of whether the delay was in the male or the female. This pattern supports the hypothesis that if the female or male of an insect delays mating, the success rate of the mating will be significantly reduced. This is different from *Cheilomenes sexmaculata* (Fabricius) (Coleoptera: Coccinellidae) [59], but similar to results reported for several other species, including *Spodoptera frugiperda* (J. E. Smith) (Lepidoptera: Noctuidae) [60], *Coccinella septempunctata* (Linnaeus) (Coleoptera: Coccinellidae) [61], *Dasylepida ishigakiensis* (Coleoptera: Scarabaeidae) [33], and *Plutella xylostella* (L.) (Lepidoptera: Plutellidae) [62,63]. This may be because as insects age, the decrease in mating success rate is caused by depletion of resources, or by the ultimate age for the eggs [64]. Although there are differences in mating systems, oviposition traits, and male–female interaction among different species, such comparisons can support our proposal to delay mating as a pest management strategy to reduce the reproductive success of the Hadda beetle.

In our results, delayed mating significantly reduced the preoviposition period. In some insect species, mating often alters female behavior by inducing physiological changes [65]. Such changes may initiate (or further stimulate) egg maturation and may be caused not only by male seminal products, but also by physical stimuli such as aedeagus insertion and the consequent stretch reception in the bursa copulatrix [65,66]. The longest pre-oviposition period was 13.17 days when the female mating delay was 0 days, while the pre-oviposition period was significantly shortened when the female mating delay was 2, 4, 6, and 8 days, and there was no difference among the treatments. This may be because the female was more mature after 2 days of mating delay, and could better retain sperm or use accessory ejaculate substances during the fertilization process [66,67], so as to further induce physiological changes and advance oviposition behavior. When the male mating delay was less than 4 days, the pre-oviposition period was significantly shortened. This may be because the mating duration was prolonged with the increase in male age, and the physiological changes of the females were caused by physical stimulation [66]. However, there was no significant difference in the pre-oviposition period between the three treatments when the male mating was delayed for 4, 6, and 8 days, which may be due to no significant change in the mating duration of the male among the three treatments.

The age of an insect at mating has been reported to be an important factor affecting its reproductive output [27,33,36,61,68]. Our results suggest that the number of days mating is delayed can have a negative influence on the reproductive performance of *H. vigintioctopunctata*, with the extent of the impact differing between the sexes. The females that did not delay mating laid more than 1000 eggs during their entire oviposition period, a result similar to that reported by Zhou et al. [17]. The number of eggs decreased as the mating delay days increased, but the decreased range was bounded and will not continue to decrease to 0. The influence of male age on female fecundity was more pronounced. Fewer eggs were laid by females when they were paired with males subjected to an 8-day delayed mating compared to females that were subjected to the 8-day delay instead of the males, indicating that the effect on reproduction was slightly smaller when delayed mating occurred in the female. In conclusion, delayed mating reduced the fecundity or fertility of *H. vigintioctopunctata* to a substantial degree (in excess of 50%), similar to the effect noted in other insect species [33,34,59,60,61,62,63,64,68,69,70]. This may be a consequence of the eggs being retained, or even resorbed into the ovaries when the females delay mating, resulting in a reduction in offspring production [27,71]. It is also possible that as the adult males age, their reproductive glands gradually degenerate, affecting the quality and quantity of sperm; thus, the amount of secretion from their reproductive glands transferred to the female during mating could be insufficient, thereby leading to the decrease in fertility [51,72,73,74,75].

The egg hatching rate declined sharply with increases in the length of the mating delay, particularly in the female mating delay treatment. In 0-day delay pairs (3-d old), 74.3% of eggs laid by mated females were fertile compared with 47.90% and 51.35% when mating was delayed for 8 days in females and males, respectively. Although the females did produce eggs in the delayed mating treatment, a portion of the eggs were infertile [50,76]. In other insects, when the hatch rates were shown to be affected as a result of delayed mating in females versus males, the results have varied widely depending on the species. For example, egg hatchability in *A. orientalis* was not affected by mating delay [35], while delaying mating in *Cheilomenes sexmaculata* [59], *Dasylepida ishigakiensis* [33], *Zygogramma bicolorata* [34], *Menochilus sexmaculatus* (Fabricius) (Coleoptera: Coccinellidae) [40], *Coccinella septempunctata* [61], *Propylea dissecta* (Mulsant) (Coleoptera: Coccinellidae) [77,78], and *Callosobruchus maculatus* (Fabricius) (Coleoptera: Bruchidae) [70] significantly affected egg fertility. However, there is an interesting phenomenon in our study that the effect of delayed mating on females decreased linearly within 4 days, but there was no significant difference among 4 days, 6 days and 8 days. This may be because when the mating delay was less than 4 days, the egg viability and/or oocyte degradation products interfere with sperm migration and/or successful egg fertilization [33,34,70], and the egg production was relatively large, but with low egg fertility. After more than 4 days of mating delay, a large number of oocytes may have been reabsorbed [79] due to the process of apoptosis [80], resulting in a decrease in egg production. Due to the similar degree of significant shortening in the pre-oviposition period, the volume of semen transferred from the male to the female was likely the same. Therefore, when the sperm were depleted or became inactive, it is basically at the end of the female oviposition period, and the number of non-viable eggs was almost the same. There was no significant difference in fertility when mating was delayed for 4, 6, and 8 days. In addition, elevation of female egg production and/or egg laying caused by seminal substances seems to be volume-dependent [66]. When male mating was delayed for 2 days, the egg hatching rate decreased significantly. This may be due to the peak of the sperm quality and quantity on the third day of eclosion (0 days of mating delay), and then the quality of the male sperm decreased, and the quantity of sperm transferred to the female was also reduced. However, there was no significant difference in egg fertility between 2 and 4 days, and 6 and 8 days after mating delay.

Mating delay has also been found to affect adult longevity in different species in a variety of ways. In some species, delayed mating can increase adult longevity [25,43,81]. This is widely considered to be a result of the reduced energy expenditure associated with reproduction [82,83]. However, Omkar et al. [34] and Bakker et al. [84] showed that mating delays may be responsible for reducing adult longevity in *Zygogramma bicolorata*, *Y. padellus* (L.) (Lepidoptera: Yponomeutidae), and *Anastrepha ludens* Loew (Diptera: Tephritidae), possibly due to males deliberately harming females in order to stimulate a higher oviposition rate; the increased activity of females, including efforts spent in locating a mate prior to mating, i.e., crawling, flying, etc., and afterwards seeking a suitable oviposition medium; and, in males, the process of ejaculation consuming a portion of their energy. However, in *H. vigintioctopunctata,* when the mating age of the adult was delayed from 3 days to 11 days, there was a weak correlation between its longevity and delayed mating days [85]. Proshold et al. [86] suggested that the relationship between the age of females at pairing and longevity is because of a change in the allocation of nutritional reserves in response to mating. The ladybugs in our study were reared under laboratory conditions. The Hadda beetles were provided with sufficient food, rich nutrition, and a relatively small range of activity space, which made them accumulate a lot of nutrients in their bodies. Therefore, the nutrients from male ejaculation were probably masked by those from the food [66], so there was no significant difference in the longevity of females among the treatments. Furthermore, males cannot transfer all their semen to females in a single mating [66], so in our experiment, mating only once will not affect the physiology of males, which will not change the longevity of males.

In some insect species, decreased reproduction with increased age at mating was strongly correlated with a limited longevity and subsequent shorter oviposition periods when adults were mated at older ages [25,35]. The length of the oviposition period decreased as delaying the mating days increased in both *H. vigintioctopunctata* treatments. Delayed mating was previously shown to decrease the oviposition period [50,59], possibly due to adults devoting a portion of their energy to survival prior to mating, and the females that mate with older adults initiate oviposition later [35,59]. Unlike most insects, however, *H. vigintioctopunctata* generally begin ovipositing between 13 and 15 days of age, regardless of when mating occurs, and is incapable of partially offsetting the decreased oviposition period due to delayed mating by extending its lifespan.

Mating success and oviposition are two fundamental factors that determine insect population growth in the next generation [76]. Our study demonstrates that a delay in mating affects each of the demographic parameters that we investigated. With an increase in mating delay days, the net reproductive rate and the intrinsic and finite rates of increase showed a decreasing pattern in females, while the population doubling time increased when the mating age exceeded 11 days. These patterns indicate that the growth potential of *H. vigintioctopunctata* populations would decrease as mating delay days increased. Our findings are similar to those obtained in other reports, indicating that delayed mating was potentially useful for controlling the population density of a species [25,27,87]. Although the effect of delayed mating on the reduction of pest population density has not been confirmed in the field, it adds content to the theoretical framework of integrating the factors affecting population dynamics.

## Figures and Tables

**Figure 1 insects-12-00629-f001:**
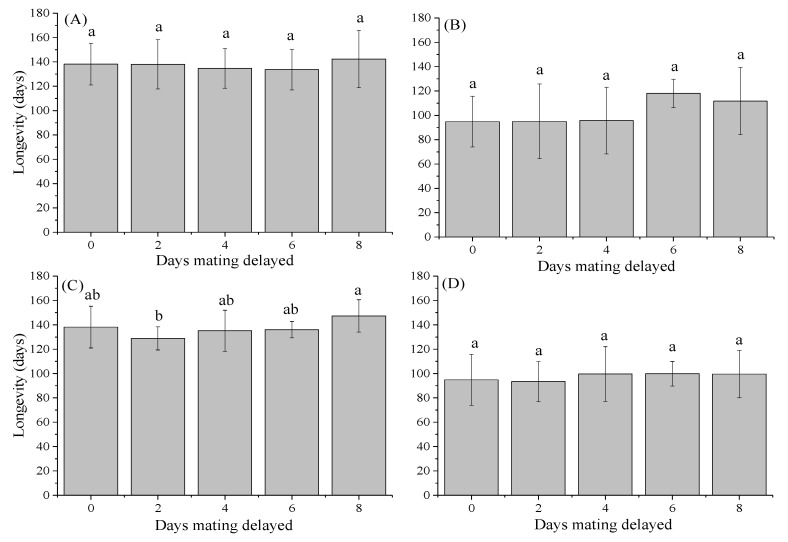
Mean (±SE) longevity of *Henosepilachna vigintioctopunctata* females after female-delayed mating (A); males after female-delayed mating (B); females after male-delayed mating (C); and males after male-delayed mating (D). Values (±SE) with the same letter are not significantly different (*p* > 0.05).

**Figure 2 insects-12-00629-f002:**
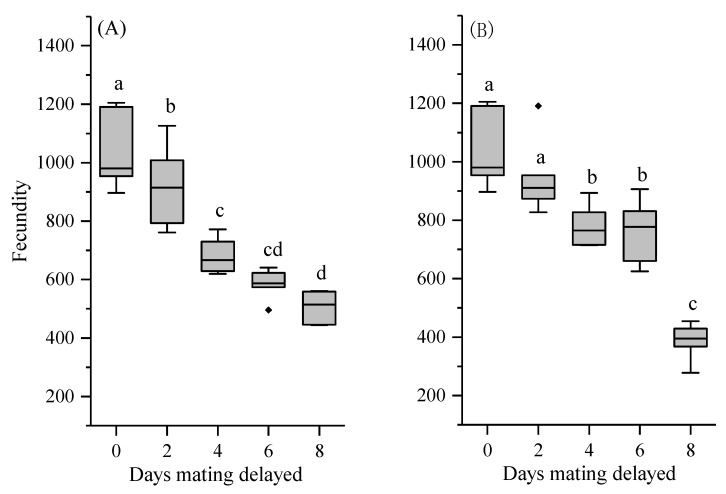
Fecundity of *Henosepilachna vigintioctopunctata* after female-delayed (**A**) and male-delayed (**B**) mating for 0–8 d. Box plots with the same letters are not significantly different (*p* > 0.05). The height of the box shows the upper and lower quartiles; the whiskers indicate the upper and lower extremes. The horizontal line in the middle of the box is the median point (half above and half below) of the data. The dots on the outside of the box indicate outliers.

**Figure 3 insects-12-00629-f003:**
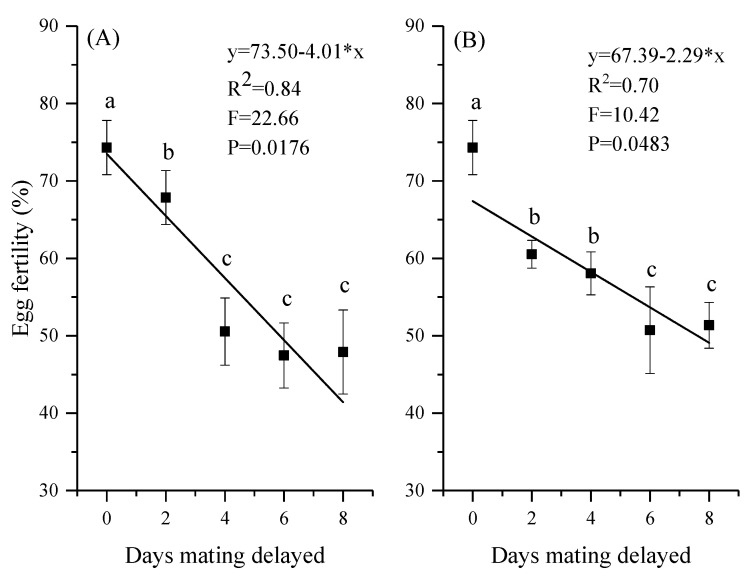
Mean (±SE) egg fertility of *Henosepilachna vigintioctopunctata* after female-delayed (**A**) and male-delayed (**B**) mating for 0–8 d. Values (±SE) with the same letter are not significantly different (*p* > 0.05).

**Table 1 insects-12-00629-t001:** Effect of delayed mating on the mating success, pre-oviposition period, and oviposition period of *H. vigintioctopunctata*.

Days Mating Delayed	Mating Success Ratio (%)		Pre-Oviposition Period (Days)		Oviposition Period (Days)	
	Female-Delayed Mating	Male-Delayed Mating	Female-Delayed Mating	Male-Delayed Mating	Female-Delayed Mating	Male-Delayed Mating
0	75.00	75.00	13.17 ± 1.17 a	13.17 ± 1.17 a	94.33 ± 13.82 a	94.33 ± 13.82 a
2	66.67	60.00	5.83 ± 1.94 b	7.33 ± 0.82 b	75.83 ± 6.74 b	81.67 ± 8.17 b
4	54.55	66.67	5.83 ± 1.47 b	5.83 ± 0.75 c	66.00 ± 7.46 b	72.00 ± 6.69 bc
6	60.00	46.15	6.00 ± 0.89 b	5.50 ± 0.84 c	64.33 ± 6.98 b	67.50 ± 6.66 c
8	50.00	42.86	4.50 ± 2.74 b	5.33 ± 0.52 c	45.33 ± 23.43 c	65.33 ± 7.06 c

Note: The data in the table are means ± SE. The same lowercase letters in the same column indicate the values are not significantly different (*p* > 0.05).

**Table 2 insects-12-00629-t002:** Some reproductive variables of *H. vigintioctopunctata* in relation to the number of female-and male-delayed mating days.

Variable	Correlation Coefficient	
	Female-Delayed Mating Days	Male-Delayed Mating Days
Mating success	−0.9058 *	−0.9096 *
Pre-oviposition period	−0.7834 *	−0.8381 *
Oviposition period	−0.9682 *	−0.9558 *
Fecundity	−0.9810 *	−0.9403 *
Hatch rate Female longevity Male longevity	−0.9217 * 0.6023 0.8118	−0.9228 * 0.1848 0.8142

Note: * Indicates that the correlation coefficient between the delayed days and observation variables is significant (*p*< 0.05), while those lacking * have no significant correlation (*p* > 0.05).

**Table 3 insects-12-00629-t003:** Effect of *H. vigintioctopunctata* female-delayed mating on the life table parameters.

Days Mating Delayed	Net Reproductive Rate (*R*_0_)	Mean Generation Time (*T*)	Intrinsic Rate of Increase (*r_m_*)	Doubling Time (*D*)	Finite Rate of Increase (*λ*)
0	463.097 ± 25.704 a	156.167 ± 7.012 ab	0.0397 ± 0.0020 a	17.686 ± 0.8983 c	1.0405 ± 0.0021 a
2	331.696 ± 21.236 b	146.833 ± 3.877 b	0.0396 ± 0.0014 a	17.589 ± 0.5643 c	1.0404 ± 0.0014 a
4	149.382 ± 8.917 c	153.167 ± 6.887 ab	0.0331 ± 0.0022 b	21.299 ± 1.1324 b	1.0337 ± 0.0023 b
6	131.615 ± 7.116 cd	154.00 ± 2.745 ab	0.0317 ± 0.0003 bc	21.905 ± 0.2342 b	1.0322 ± 0.0003 bc
8	95.620 ± 5.914 d	165.333 ± 5.426 a	0.0277 ± 0.0010 c	25.200 ± 0.8348 a	1.0281 ± 0.0010 c
Correlation coefficient	−0.9406 *	0.6025	−0.9663 *	0.9575 *	−0.9666 *

Note: The data in the table report the days mating was delayed, as the means ± SE. Means with the same lowercase letter within a column are not significantly different (*p* > 0.05). * Indicates the correlation coefficient between the delayed days and observed variables is significant (*p* < 0.05), and those correlations lacking * are not significant (*p* > 0.05).

**Table 4 insects-12-00629-t004:** Effect of *H. vigintioctopunctata* male-delayed mating on the life table parameters.

Days Mating Delayed	Net Reproductive Rate (*R*_0_)	Mean Generation Time (*T*)	Intrinsic Rate of Increase (*r_m_*)	Doubling Time (*D*)	Finite Rate of Increase (*λ*)
0	463.097 ± 25.704 a	156.167 ± 7.012 a	0.0397 ± 0.0020 a	17.685 ± 0.8981 b	1.0405 ± 0.0021 a
2	282.662 ± 15.758 b	156.000 ± 8.266 a	0.0367 ± 0.0020 ab	19.188 ± 1.039 b	1.0374 ± 0.0021 ab
4	247.203 ± 8.569 b	152.667 ± 6.682 a	0.0365 ± 0.0018 ab	19.233 ± 0.907 b	1.0372 ± 0.0019 ab
6	148.633 ± 9.897 c	151.667 ± 6.800 a	0.0333 ± 0.0019 b	21.122 ± 1.104 b	1.0339 ± 0.0019 b
8	67.606 ± 4.235 d	160.333 ± 9.649 a	0.0269 ± 0.0022 c	26.527 ± 1.811 a	1.0272 ± 0.0022 c
Correlation coefficient	−0.9767 *	0.1851	−0.9408 *	0.8987 *	−0.9397 *

Note: The data in the table report the days mating was delayed, as the means ± SE. Means with the same lowercase letters within a column are not significantly different (*p* > 0.05). * Indicates the correlation coefficient between the delayed days and observed variables is significant (*p* < 0.05), and those correlations lacking * are not significant (*p* > 0.05).

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
