# Peer review of "Effects of Delayed Mating on the Reproductive Performance of Henosepilachna vigintioctopunctata (F.) (Coleoptera: Coccinellidae)"

_insects, 2021, doi:10.3390/insects12070629_

Round 1
Reviewer 1 Report
During the peer review process, I found the following reference by the authors.
It is a Chinese journal called "Entomological Knowledge". I found the abstract in English, so I didn't have the name of the insects used as material, but the content of their research seems to be the same with this manuscript.
There is no citation in this paper, so I don't know the relationship between this paper and the one already published. In addition, if the material is the same, it will lose originality and there is a fear of double submission.
Please give me the details of the following paper so that I can review it correctly. If possible, please send me a PDF of the original paper. Also, please tell us the name of the material.
《Entomological Knowledge》 2004-04
"Effect of delayed mating on the insect procreation behavior and its relationship with the pheromone control method"
WANG Xiang-Ping,ZHANG Zhong-Ning ** (State Key Laboratory of Integrated Management of Pest Insects and Rodents, Institute of Zoology, Chinese Academy of Sciences, Beijing 100080, China)
Abstract: Effects of insect delay mating on the fecundity, egg fertility, mating success, and longevity are reviewed. When male mating was delayed, the fecundity, egg fertility, oviposition period of female decreased, while sperm quality and longevity of male usually increased. The relationship between the delay mating and pheromone control is discussed.
Yours sincerely
Author Response
Response: Thank you for your comments on our manuscript. "Effect of delayed mating on the insect procreation behavior and its relationship with the pheromone control method" is a review published by Wang Xiang-Ping and Zhang Zhong-Ning in the Entomological Knowledge. In the review, Wang and Zhang summarized the effects of delayed mating on mating behavior and reproductive performance of a variety of Lepidoptera insects, such as Helicoverpa armigera, Bupalus piniarius, Cryptophlebia illepida, Ephestia cautella, Spodoptera exigua and so on. The effects of delayed mating on different insect species are different. Now, we are studying the effect of delayed mating on the Coleoptera, Henosepilachna vigintioctopunctata. I will send you a PDF of Wang and Zhang's original paper. Please see the attachment.
We would like to thank the referee again for taking the time to review our manuscript.

Reviewer 2 Report
The manuscript “Effects of delayed mating on the reproductive performance of Henosepilachna vigintioctopunctata (F.) (Coleoptera: Coccinelidae)” reports results from a laboratory experiment testing the effect of delaying mating on reproduction of an herbivore ladybird beetle. Four mating delays were applied to females, and also to males. Authors generally show that delaying mating negatively affects the reproductive capacity of H. vigintioctopunctata, but not its longevity. The results provide fundamental knowledge that could be used to develop control techniques in fields to control this crop herbivore.
The study provides potential important and useful results to improve our knowledge on this species of ladybird beetle. I have nonetheless some concerns regarding the experiments and comments on the manuscript:
General comments:
- How many individuals were finally used in each treatment? In the M&M, it is indicated that 15 replicates are used for each treatment. But what is the effect of mating success on this number of replicates? When mating is delayed by 8 days, mating success is said to be 40 to 50% (Table 1). Does it mean that only 40 to 50% of the replicates are finally used (i.e., 6 to 7 individuals)? Moreover, a minimum of 30 replicates per treatment is recommended.
- There is a potential confusion between mating delay and the age of the individuals at mating. Mating delay starts from an age of 3 days old as explained L. 113-117. But a male and a female do not necessarily mate directly after being paired. They may potentially be in the same Petri dish without mating for some days. In this case, the mating age is not equal to the “pairing age”, here being “mating delay + 3 days”. But in the manuscript, it seems that “mating age” and “days of delayed mating” are considered equal (e.g., L. 172 (“adults’ age”), 175, 238-241, 276, 336).
- Pre-oviposition period is defined here as “number of days from after mating to the first egg laid”. I think that it is wrong. Pre-oviposition period is the time from adult emergence to first egg laid. Therefore, pre-oviposition period of females which mating was delayed by, for instance, 8 days is: 3+8+4.5=15.5 days, not very different from females which mating was not delayed (3+13=16 days). The same calculation can be done for all treatments. Effect of mating delay on pre-oviposition period (i.e., time from emergence to first oviposition) may not be significant.
- What is the relation between longevity and oviposition period?
- Daily fecundity can help disentangle effect of longevity and oviposition period on total egg production. Daily fecundity can be of significant importance since mortality can occur at any time in fields notably because of predators.
- With regards to statistical analyses:
-- Generalized Linear Mixed Models were fitted. But what are the random effects?
-- Which tests were conducted on the fitted models? Fitting GLMM does not provide F-values. In addition, likelihood-ratio tests are more appropriate that F-test for GLMM with binomial distribution.
-- Poisson error-distribution or negative binomial distribution (in case of data overdispersion) are more appropriate than normal distribution for counting data.
-- How was performed the linear regression showed in Fig. 2?
-- To me, the results provided by Person correlation do not provide anything more than the results provided by the GLMM (the trend of effects are obviously the same); and using GLMM is much more appropriate than using Pearson correlation. To me, Pearson correlation tests are not needed.
- Results are discussed with regards to many other different insect species. But none of them are other Coccinellidae species. Same comment applies for the Introduction. Comparing the present results to other Coccinellidae species is expected, as the reproductive patterns are potentially more related.
More line-by-line comments:
L.3: Coccinellidae (and not Coccinelidae).
L.42: “et al.” must be deleted here.
L.50: “a large number of generations per year”. Please be more specific.
L.51-54: Please provide more specific information here.
L.55-68: It is said that management techniques are varied but only chemical control is presented. What are the other techniques?
L.103: How many larvae were collected from the field?
L.106: Was it the second lab generation (F2) that was used in the experiment? Please clarify.
L.128-129: The sentence into brackets is not clear to me. Please rephrase.
L.167 (and also L. 220, 274): What is “normal”?
L.169-171: This sentence is confusing. Please rephrase.
L.193: “extremely low” is not needed here. Whether it is extremely low or not can be a relative perception. “significantly lower” is more accurate.
L.198: Results on longevity could be provided earlier.
L.227-228: Effect is not significant at 0.05% (P=0.2304), contrarily to what is said.
L.316: A reference is needed here after “not affected”.
Reviewer 3 Report
Insects 1237978
The authors demonstrate that when mating is delayed, these insects will have a decrease in egg output which results in decreases in demographic parameters. The authors suggest that the impact on fecundity could make these insects a good candidate for mating disruption as a pest management technique. The authors do a good job with the introduction and set up of the reason for these experiments and they provide good discussion as to their results.
Specific comments
Lines 26-28: The statement about preoviposition period is confusing. You say the period is reduced by mating delays, but then not significantly impacted by increasing age. Seems to not match up.
Line 120: Virgin beetles were randomly selected? Did you age them to 3 days to begin as well, or were some 10 days old? I'm not sure that this is a good enough description of how you got your virgin beetles. They should all be aged appropriately, not randomly.
Lines 169-171: Are you comparing male delays with female delays here? I'm not sure what result you're trying to say.
Figure 1. Please indicate what the boxes are representing and the lines within the boxes? Median or mean? What do the dots mean on the graph? Outliers?
Figure 2. The equation, R2, F, and p information is too clustered together. I would try to stack it or at least separate it with a semi-colon. Are your error bars 95% intervals?
Figure 3. Where are the SE values on the graph? I think the axis with the Longevity labels needs to be cleaned up. It just runs together and is very hard to read. You could also just make this a flat graph and have longevity on the y-axis with each longevity measured and days delayed grouped. Also, are the letters for significance for the entire graph or just for each longevity group?
Table 2. I think you should still provide the correlation coefficient for Female and male longevity but just indicate they are not significant by not putting the asterisk.
Lines 218-221: What is the normal net reproductive rate value?
Table 3. I think you should report the correlation coefficient for mean generation time as well. Something with the formatting of this table needs to be corrected so that significance letters are on the same line all the time.
Table 4. Same comments as table 3.
Lines 261-264: In a controlled experiment like this, the decrease in reproduction capacity is not likely due to decreased attraction as they do not need to actually find their mates. It's more likely due to a metabolic or biological depeltion of resources or the ultimate age fo the eggs. I would mention this, as you didn't really test for any sort of attraction component in this paper.
Round 2
Reviewer 1 Report
This study examined the effects of delayed mating in Henosepilachna vigintioctopunctata from both the male and female perspective, and found some very interesting results.
I think it's worthy of being published in “Insects”.
However, before doing so, here are some points that I think should be corrected.
L11. ‘which feeds on plants,’ Delete.
L15. The connection from ‘biological pest control’ to ‘delay mating’ is abrupt. It is difficult to understand the connection.
L16. ‘of delayed mating’ Delete. The following lines describe ‘delayed mating’.
L22. The 'population growth parameters' are ambiguous and difficult to understand.
L22. Beginning three days after emergence for both sexes,
L23-25. We compared the data when mating was delayed for males only with the data when mating was similarly delayed for females only.
L30. What is ‘life table parameter’?
L39. the 28-spotted potato ladybird or the Hadda beetle. Use either of them. ‘the ladybird beetle’ is not suitable for Henosepilachna vigintioctopunctata.
L40. Include Australia and Brazil, as mentioned in ref.9.
L40-41. Both larvae and adults feed mainly on the leaves of Solanaceae plants,
L55. Host plant species? or parts?
L56-58. Why is it so difficult to control them when they feed and lay their eggs in different places? Isn't it difficult because it is controlled on a field or district basis?
L60-62. This sentence is redundant and doesn't explain much of what you want to say on this paper. The difficulty of pest control of this species is well described in the sentences after L63, so I think it can be deleted.
L94-96. Isn't that already explained in the above paragraph? It is a duplicate.
L11-115. Please explain the rearing method in more detail. Size of the case, temperature conditions, etc.
L122. How did you separate by sex?
L134-135. What is copulation duration from insertion to ejection? Also, why is complete mating 30 minutes? Perhaps it is the time it takes for sperm to enter, but what is the rationale?
L139. How were the collected eggs reared? How did you examine the hatching rate?
L186-187. How did you collect this data? Is it described in the methods?
L191. Figure 1. The two data on the left should be the same, but why is the letter different (ab and a)?
Discussion
The authors reviewed other species and compared their results with those of this species, but it would be better to discuss why this species is different from others, and to compare the ecological characteristics of this species with others.
I think that each sex has different adaptive and investment strategies due to delayed mating. Because the costs and benefits are different for each female and male. Here, it would be better to discuss more carefully how each sex changes its strategy in its own age and in the age of its mating partner. Also, it is understandable that the physiological deterioration of the female due to her age would cause her egg production to decrease, but please discuss more carefully whether the male's age difference of just a few days would cause his mate's egg production to decrease, including other citations such as sperm deterioration, ejaculate deterioration, etc.
Why does delayed mating significantly reduce the number of eggs laid while lifespan remains the same? Didn't the egg laying rate decrease?
L261. 25-32%
L263. What is this patter? Both sexes? Day of the delayed mating (8 day)? Reduction rate?
L268-271. The cause may be age as well, but many things are different in different species, such as mating system, egg laying traits, male/female interactions, etc., so do you really need to compare them in detail here? Also, what kind of knowledge value do you expect to get from it?
L272-277. It's better to discuss the two sexes separately.
L331. What is ‘limited longevity’?

Reviewer 3 Report
Much improved. An overall editorial read through to catch some minor errors is needed.
Author Response
Dear Reviewer,
Thank you very much for reviewing our manuscript. Thanks for your help, our manuscript is more perfect. Thank you for your valuable advice. This time, we read an overall editorial and revised some minor errors, such as the font and paragraph arrangement, in the manuscript.
Thank you very much.
Kind regards,
Yaling Wang
Round 3
Reviewer 1 Report
The authors have responded very carefully to my previous comments.
I have some comments on the revised version.
The results are very interesting and worthy of publication, but there is still some revision needed in the discussion. Please refer to the following remarks and do your best to prepare a better paper.
L24. The authors replaced ‘population life table parameters’ with ‘population growth parameters’. But 'population life table parameters' are also ambiguous and difficult to understand. Please be a little more detailed in describing the subjects you investigated.
L30. What is ‘life table parameter’?
Response: Thank you very much. In our manuscript, the life table parameters refer to the net reproductive rate, intrinsic and finite rates of increase, doubling time and finite rate of increase in the time specific life table.
Please state it in the text.
L44. is widely distributed in Australia(ref.),
Methods: There are many unclear points about the method in general. Please explain carefully and don't omit too much.
L119-120. What I want to ask is not the temperature range that the artificial weather equipment can be set to, but what temperature, what photoperiod, and what percentage of humidity the beetles were actually kept at. I also want to know about the plastic cases and other containers you used to keep the beetles in. Also, the website referring RXZ-280 says that the capacity is 300 liters, but is it correct that it is 288 liters?
L122. How did you separate by sex?
Response: Thank you very much. The abdomen of the adult was observed under the microscope. There was a longitudinal indentation at the end of the abdomen of the female, while the male was flat and smooth without indentation. The identification accuracy was 100%.
Briefly describe or refer to the method of sexting, please.
L130. the adult females and males did not become sexually active until 3 days of age (ref.),
L135. What is copulation duration from insertion to ejection? Also, why is complete mating 30 minutes? Perhaps it is the time it takes for sperm to enter, but what is the rationale?
Response: Thank you very much. In our experiment, the shortest mating time of the Hadda beetle was no more than 10 minutes, and the longest was 138 min, but mainly concentrated in 30 70 min. Combined with the previous work experience in the laboratory, the Hadda beetle needs at least 30 min of mating to produce the Hadda beetle needs at least 30 min of mating to produce fertile eggs. Therefore, we regarded the mating duration of more than 30 min as the completion of mating in this study.
- 140-141. Please state it in the text. There is a lack of explanation about mating behavior, so please explain it in more detail.
L139. How were the collected eggs reared? How did you examine the hatching rate?
Response: Thank you very much. Beetles laid their eggs on the leaves of Solanum nigrum or on the surface of filter paper in the oviposition cages. When collected eggs every day, we only need to take out the leaves and filter paper with egg mass. After the daily egg production of the female was determined under the microscope, the egg mass of each female was separately placed in a sealed culture dish. After the eggs hatched, the number of larvae or non hatched eggs was counted to obtain the corresponding hatching amount. After the experiment, the hatching rate of each female was calculated.
L144-147. Please briefly describe the above in the text.
L191. Figure 1. The two data on the left should be the same, but why is the letter different (ab and a)?
Response: Thank you very much. The same data was analyzed in two combinations of female only delayed and male only delayed. In these two combinations, the change degree of female longevity in each treatment was different, so the significant difference level of female longevity without delayed mating was also different.
You mean males are compared to males, females are compared to females, and not between males and females, right? If so, you need to change the symbols for males and females, for example, by separating them into upper and lower case. If it's the same `a`, the results was statistically processed and there was no significant difference.
L276-277. You might want to explain it more carefully here. Since it was carefully explained in the cover letter, please state it carefully in the text as well, including references.
L278-281. This explanation does not explain why there is no significant difference between a 2-day delay and an 8-day delay; isn't there some other explanation for why even a 2-day delay results in a significantly shorter preoviposition period than a 0-day? Similarly, I don't think the results for the males have been adequately explained or discussed. Please reconsider.
L285-286. The data in Fig.1 show that lifespan does not change, yet this discussion states that there is a trade-off between reproduction and lifespan, which I do not understand. Isn't the significant change in the pre-oviposition period a physiological response to even a small delay in mating?
L278-287. In response to the fact that delayed mating significantly alters the pre-oviposition period, it is important to discuss the issue from the perspective of females and males separately. The physiological responses of females can be discussed by referring to the literature on insect physiology, and males can be discussed by referring to changes in ejaculate or the results of cryptic female choice. With regard to the results of changes in the fecundity, egg fertility and pre-oviposition period, it has been reported that they vary significantly with the amount of ejaculate (mating time). This paper may also be useful for discussion. Himuro and Fujisaki (2015). Effects of mating duration on female reproductive traits of the
seed bug Togo hemipterus (Heteroptera: Lygaeidae). Appl. Entomol. Zool. DOI 10.1007/s13355-015-0357-4
L289-290. The females that did not delay mating laid more than 1,000 eggs during their entire oviposition period,
L294. ‘although the decrease was limited’. I still don't understand the meaning of this sentence. Please explain it carefully.
L307-. The results in Fig.3 are very interesting: the effect of delayed mating on females is linearly decreasing up to 4 days, but there is no significant difference between 4, 6 and 8 days. In other words, it is an asymptotic line. It is interesting to note that even if there is a drop, fertility can be ensured up to around 50%. You should discuss the changes more. Also, it is important to discuss the issue from the perspective of females and males separately.
L315-319. Even in many of the references you cited, I don't know whether the citation is for a male or female effect, or for either phenomenon. Again, it is important to discuss the issue from the perspective of females and males separately.
L336-339. It is difficult to understand what you are trying to say about the discussion of the connection between sex differences in life span and delayed mating.
